# Molten Salt Corrosion Behavior of Dual-Phase High Entropy Alloy for Concentrating Solar Power Systems

**DOI:** 10.3390/e25020296

**Published:** 2023-02-04

**Authors:** Kunjal Patel, Vahid Hasannaeimi, Maryam Sadeghilaridjani, Saideep Muskeri, Chaitanya Mahajan, Sundeep Mukherjee

**Affiliations:** Department of Material Science and Engineering, University of North Texas, Denton, TX 76203, USA

**Keywords:** molten salt corrosion, scanning kelvin probe (SKP), work function, galvanic corrosion, high entropy alloy, dual-phase alloy

## Abstract

Dual-phase high entropy alloys have recently attracted widespread attention as advanced structural materials due to their unique microstructure, excellent mechanical properties, and corrosion resistance. However, their molten salt corrosion behavior has not been reported, which is critical in evaluating their application merit in the areas of concentrating solar power and nuclear energy. Here, the molten salt corrosion behavior of AlCoCrFeNi_2.1_ eutectic high-entropy alloy (EHEA) was evaluated in molten NaCl-KCl-MgCl_2_ salt at 450 °C and 650 °C in comparison to conventional duplex stainless steel 2205 (DS2205). The EHEA showed a significantly lower corrosion rate of ~1 mm/year at 450 °C compared to ~8 mm/year for DS2205. Similarly, EHEA showed a lower corrosion rate of ~9 mm/year at 650 °C compared to ~20 mm/year for DS2205. There was selective dissolution of the body-centered cubic phase in both the alloys, B2 in AlCoCrFeNi_2.1_ and α-Ferrite in DS2205. This was attributed to micro-galvanic coupling between the two phases in each alloy that was measured in terms of Volta potential difference using a scanning kelvin probe. Additionally, the work function increased with increasing temperature for AlCoCrFeNi_2.1,_ indicating that the FCC-L1_2_ phase acted as a barrier against further oxidation and protected the underlying BCC-B2 phase with enrichment of noble elements in the protective surface layer.

## 1. Introduction

The world economy is highly dependent on fossil fuels to meet persistent energy demands, which results in carbon emissions and negative environmental impacts. This is driving the global search for sustainable and alternative energy sources [1], solar energy being one of the most viable and clean options [2]. The global capacity for concentrating solar power (CSP) systems has increased fifteen-fold over the last two decades [3]. The solar power conversion systems in CSP operate with advanced fluids above 400 °C [4], using molten salts such as solar salt [5], Hitec salt [6], and MgNaK chlorides [7]. In particular, the ternary MgNaK chloride (MgCl_2_-NaCl-KCl) eutectic salt mixture has high thermal stability and is more cost effective compared to other molten salts [8,9]. This allows CSP to operate at higher temperatures for higher efficiency of thermal-to-electrical energy conversion and reduces the cost of electricity. However, degradation of structural materials in a molten salt environment remains a critical concern in CSP systems [9,10]. The high operating temperature and reactive nature of the salt significantly increase the corrosion susceptibility of the materials and reduce component lifetimes [4]. This necessitates the development of new alloys with high-temperature strength and corrosion resistance to withstand the harsh molten salt environment of CSP systems.

High-entropy alloys (HEAs) with multiple principal elements in (nearly) equimolar proportions [11,12] have attracted much attention recently because the unconventional alloying approach results in excellent mechanical properties and surface degradation resistance [13,14,15,16,17,18]. Most of the reported HEA systems have a single-phase body-centered cubic (BCC) or face-centered cubic (FCC) structure [19,20,21]. To achieve a good balance of high strength of BCC HEAs along with good ductility of FCC HEAs, eutectic high-entropy alloys (EHEAs) consisting of the dual phases in lamellar microstructure have been recently developed [22,23]. Some examples of EHEAs with excellent combination of strength and ductility include CoCrFeNiZr_0.5_, CoCrFeNiMnPd_1.4_, CoCrFeNiMnPd_0.6_, Nb_25_Sc_25_Ti_25_Zr_25_, AlCrFeNiMo_0.2_, Al_1.2_CrCuFeNi_2_ and AlCoCrFeNi_x_ [24]. In particular, the AlCoCrFeNi_2.1_ EHEA with lamellar arrangement of BCC (B2) and FCC (L1_2_) solid-solution phases has shown the most promising mechanical properties [22,25,26,27]. AlCoCrFeNi_2.1_ EHEA showed superior pitting corrosion resistance compared to 304 stainless steel in chloride solution at room temperature [27]. This EHEA exhibited an excellent combination of high yield strength (~750 MPa) and ductility (elongation ~18%) at room temperature [22,28]. However, the molten salt corrosion behavior of dual phase HEAs for potential use in CSP systems has not been reported so far, and fundamental understanding of the underlying high-temperature corrosion mechanisms is lacking.

Here, the corrosion behavior of AlCoCrFeNI_2.1_ EHEA in molten NaCl-KCl-MgCl_2_ salt was investigated at 450 °C and 650 °C. The performance of EHEA was compared with one of the benchmark dual-phase structural alloys, namely duplex steel 2205 (DS2205). Dual-phase steels have been reported to have excellent corrosion resistance and a good combination of strength and ductility [29,30,31,32,33,34,35]. To explain the effect of eutectic microstructure on the local electrochemical activity, Scanning Kelvin Probe (SKP) analysis was utilized to measure the relative electron work function [27,36,37]. Our findings pave the way for further research on molten salt corrosion behavior of dual-phase alloys over a wide range of temperature for potential use in concentrating solar power systems.

## 2. Materials and Methods

The AlCoCrFeNi_2.1_ eutectic high-entropy alloy ingot was prepared by vacuum-arc-melting high purity constituent elements (with 99.95 wt.% elemental purity). The ingots were re-melted multiple times for homogeneity. Samples with dimensions 20 mm × 5 mm × 2 mm were cut from the alloy for counter and working electrodes. The samples were polished to 1200-grit silicon carbide paper, rinsed with distilled water, ultrasonically degreased with acetone, and dried. A schematic and experimental setup of the furnace using three electrodes for the electrochemical studies are shown in Figure 1a,b. The custom-built tube furnace consisted of three electrodes submerged up to 5 mm in the eutectic salt mixture in open air at the two temperatures of 450 °C and 650 °C. The electrochemical measurements were performed using a potentiostat (Reference 3000, Gamry, Warminster, PA, USA). A Pt sheet (15 mm × 5 mm × 0.5 mm) in a quartz tube was used as the reference electrode (RE). A porous membrane made of alumina was used at the bottom of the quartz tube to provide ionic conductivity in the molten salt. The porous membrane minimizes salt migration and provides good electrical continuity. To eliminate any mixed potentials in the three-electrode system, the counter electrode was the same alloy as the working electrode in all experiments. The surface area ratio of the counter electrode (CE) to the working electrode (WE) was kept fixed at 1 (CE:WE = 1:1) [38]. The working and counter electrodes were placed in open-ended quartz tubes to electrically isolate them during experiments.

Analytical grade MgCl_2_, NaCl, and KCl (Sigma Aldrich, St. Louis, MO, USA) were used to prepare the eutectic salt mixture in the proportion of 45.4 wt.%, 33 wt.%, and 21.6 wt.% respectively. Melting point of the salt mixture was determined experimentally to be ~380 °C, which is consistent with the value reported in the literature [39]. One application is in indirect storage system with molten salt as storage medium, often used in a parabolic trough CSP plant with maximum temperatures in the range of around 400–450 °C. Another configuration is direct storage system, often used in a tower CSP plant with molten salt heat transfer fluid (HTF), as well as TES with maximum temperatures in the range of around 600–650 °C [4,5,6,7]. The eutectic salt mixture was added in an alumina crucible to obtain a salt melt depth of ~20 mm. The NaCl-KCl-MgCl_2_ salt was purified by annealing at 300 °C for 24 h in high purity Ar atmosphere to remove moisture and other impurities [39]. Inductively coupled plasma-optical emission spectroscopy (ICP-OES) was used to determine impurity concentration in the eutectic salt mixture before the corrosion test as summarized in Table 1. After purification, the residual moisture in the salt is expected to be low. However, a detailed evaluation of the effect of moisture content on corrosion behavior has not been performed in the current study, but is part of a future study.

Open circuit potential (OCP) and potentiodynamic polarization tests were conducted for each alloy. The OCP was recorded for at least 1 h after immersion to reach a stable value before the potentiodynamic polarization tests. The potentiodynamic tests were performed at a scan rate of 1 mV/s in the potential range of −0.5 to 1.5 V with respect to Pt pseudo-reference electrode. Three tests were performed for each condition. 

Scanning electron microscopy (SEM, FEI Quanta-ESEM 200) equipped with energy dispersive spectroscopy (EDS) and electron backscatter diffraction (EBSD) were performed to analyze the microstructure and volume fraction of phases present in the alloys. The crystal structure of the alloys and corrosion products were determined by X-ray diffraction (XRD) using a Rigaku Ultima X-ray diffractometer with 1.54 Å Cu-Kα radiation and a scattering angle in the range of 20° to 90°. A scanning kelvin probe (SKP) microscope (Princeton Applied Research) was used to measure the relative work function distribution or Volta potential over the surface. A tungsten wire with a perpendicular amplitude of 30 µm with respect to the samples at a frequency of 80 Hz was used as a reference probe during SKP measurements. Work function was determined at a tip to sample separation of 50 µm in lab air (RH 55%).

## 3. Results

### 3.1. Microstructural Characterization

The as-cast microstructures of AlCoCrFeNi_2.1_ and DS2205 are shown in Figure 2a–f. The microstructure of AlCoCrFeNi_2.1_ consisted of lamellar arrangement of FCC (L1_2_—light contrast) and BCC (B2—dark contrast) solid solution phases (Figure 2a) with an average grain size of ~75 ± 25 µm. Similarly, DS2205 showed a lamellar morphology of FCC (γ austenite—light contrast) and BCC (α ferrite—dark contrast) phases (Figure 2e) with an average grain size of ~110 ± 35 µm. The EBSD phase map in Figure 2b shows the lamellar structure of AlCoCrFeNi_2.1_ with FCC (red color ~77%) and BCC (green color ~23%) phases, while Figure 2f shows the corresponding map for DS2205. The EDS map of the as-cast AlCoCrFeNi_2.1_ is shown in Figure 2c. The BCC phase was found to be rich in Ni and Al, while the FCC phase was rich in Co, Cr, and Fe. Similarly, the EDS map of DS2205 in Figure 2g shows that the FCC (γ—Austenite) phase is rich in Ni and the BCC (α—Ferrite) phase is rich in Cr and Mo. The XRD pattern in Figure 2d of as-cast AlCoCrFeNi_2.1_ confirmed its dual-phase microstructure consisting of FCC-L1_2_ and BCC-B2 phases. Corresponding XRD pattern for DS2205 in Figure 2h confirmed its dual-phase FCC-BCC microstructure. The chemical composition of each phase in AlCoCrFeNi_2.1_ and DS2205, as well as the overall composition, are summarized in Table 2. The differences in the fraction of relatively nobler elements (e.g., Co, Cr, and Ni) between the two phases in AlCoCrFeNi_2.1_ is significantly larger than in DS2205. 

### 3.2. Electrochemical and Corrosion Behavior

Open circuit potential for the two alloys at 450 °C and 650 °C is shown in Figure 3a. DS2205 and AlCoCrFeNi_2.1_ showed stable potential, which gradually increased towards a more noble value and lower fluctuations after ~1 h of exposure in the molten chloride salt at 450 °C. At 650 °C, both the alloys showed relatively poor stability for the first 30 min, suggesting that an increase in temperature might have resulted in the unstable growth of surface oxide initially, before reaching steady state. Figure 3b shows the potentiodynamic polarization curves for the two alloys after stable OCP at the temperatures of 450 °C and 650 °C. The corrosion current density (I_corr_) for the two alloys was determined from Tafel analysis and is summarized in Table 3. The I_corr_ value increased with an increase in temperature for both alloys, indicating the high dissolution rate and relatively poor protective nature of their surface passivation layer. AlCoCrFeNi_2.1_ showed lower I_corr_ at 650 °C compared to DS2205, demonstrating its excellent corrosion resistance at the higher temperature. 

Corrosion rates (CR) based on Icorr and Mass loss for the two alloys was calculated from the following equations [36,40]:(1)$$CR\;\left(\frac{\mathrm{mm}}{\mathrm{year}}\right)=K_{1}*\frac{I_{corr}\times EW}{\rho }$$
where K_1_ = 3.27 × 10^−3^ [mm/(μA·cm·year)] is a constant which is taken from ASTM G102; EW is equivalent weight in g/equivalent; *ρ* is density in g/cm^3^; and I_corr_ is corrosion current density in μA/cm^2^.
(2)$$CR=\frac{k.m_{loss}}{A.t.\rho }$$
where k is constant 8.76 × 10^4^; CR is in (mm/y), which is taken from ASTM G31; m_loss_ is the mass loss (g) of the metal in time t (h); A is the surface area of the material exposed (cm^2^); and *ρ* is the density of the material (g/cm^3^). The corrosion rate based on Equations (1) and (2) for the two alloys at 450 °C and 650 °C are shown in Table 3. AlCoCrFeNi_2.1_ showed a significantly lower corrosion rate compared to DS2205 at both the temperatures. Poor stability of Fe-based alloys at elevated temperatures in molten chloride media due to dissolution of Cr and Fe has been reported previously [38,41]. In contrast, AlCoCrFeNi_2.1_ high-entropy alloy showed a higher corrosion resistance compared to dual-phase DS2205 steel possibly, due to greater stability of its constituent elements in the molten chloride salt environment. 

Mass loss measurements based on immersion study was performed for determining material loss due to corrosion. Figure 4 shows the mass loss (mg/cm^2^) of the EHEA and DS2205 alloys after immersion in NaCl-KCl-MgCl_2_ at 450 °C and 650 °C for 24 h. Greater material loss was observed with an increase in temperature, which supports the potentiodynamic study (Table 3). At 450 °C, mass loss for DS2205 was ~24 mg/cm^2^ after 24 h, which was slightly higher than AlCoCrFeNi_2.1_ (~19 mg/cm^2^ after 24 h). At 650 °C, mass loss drastically increased for DS2205 to ~69 mg/cm^2^ after 24 h compared to ~38 mg/cm^2^ for AlCoCrFeNi_2.1_. Figure 4 shows the lower mass loss for AlCoCrFeNi_2.1_ as compared with DS2205 at both temperatures, suggesting superior corrosion resistance of the EHEA. 

Figure 5a,b show the optical microscopy images of AlCoCrFeNi_2.1_ and DS2205 after corrosion tests at 450 °C and 650 °C, respectively. SEM images of the surfaces after corrosion are shown in Figure 5c–f. Selective dissolution of the B2 phase in AlCoCrFeNi_2.1_ was observed at 450 °C (Figure 5c). DS2205 showed preferential pitting in the BCC (α—Ferrite) regions while the FCC (γ—Austenite) phase was relatively intact (Figure 5d). At 650 °C, uniform pitting was observed all over the surface for AlCoCrFeNi_2.1_ (Figure 5e) in contrast to severe material loss from selective removal of the BCC (α—Ferrite) phase in DS2205 (Figure 5f). Previous studies suggest that the relative chemistry difference and phase volume fraction (FCC:BCC) determine the degree of degree of galvanic coupling in dual-phase alloys [27,42,43,44,45]. At 650 °C, duplex steel has been reported to show chloride-induced severe corrosion due to preferential attack of its ferrite phase [46]. The high-magnification SEM images in Figure 5c–f demonstrate the galvanic coupling in both AlCoCrFeNi_2.1_ and DS2205, with dual-phase microstructure, albeit to different extents.

## 4. Discussions

### 4.1. Effect of Alloy Composition on Molten Salt Corrosion Behavior

The cross-sectional morphology and element distribution of the AlCoCrFeNi_2.1_ and DS2205 samples are shown in Figure 6a–d by the EDS elemental maps, after immersion in molten salt at both the temperatures. For both the alloys, there was selective dissolution of some alloying elements and selective oxidation of BCC phase (B2—AlCoCrFeNi_2.1_ and α Ferrite—DS2205) near the surface at 450 °C and 650 °C (Figure 6). The BCC phase oxidized faster than FCC, suggesting that the BCC phase acted as micro-anodic sites with respect to the FCC phase. In AlCoCrFeNi_2.1_, it is noted that at 650 °C the concentration of Al, Cr, and to a lesser extent, Fe decreased (depletion), while the concentration of Co and Ni increased (enrichment). Cr depletion has been reported to be the primary form of material loss in Fe-based alloys exposed to chloride-environment [41,47]. In addition, Al loss is expected from AlCoCrFeNi_2.1_ because of greater stability of AlCl_3_ in the molten salt. At 450 °C and 650 °C, it is apparent from EDS maps shown in Figure 6 that Al and Cr are depleted from AlCoCrFeNi_2.1_. Although both AlCoCrFeNi_2.1_ and DS2205 alloys have similar Cr concertation (~20 at. %), DS2205 showed greater mass loss at both temperatures compared to AlCoCrFeNi_2.1_. This may be attributed to the presence of Al in AlCoCrFeNi_2.1_ that hinders the dissolution of the other alloy constituents in the molten salt [20]. In particular, the Cr depletion was significantly greater for DS2205 compared to AlCoCrFeNi_2.1_. At 450 °C, the Cr depletion depth in in AlCoCrFeNi_2.1_ was ~30 μm_,_ while that for DS2205 was ~200 μm_._ Cr depletion depth increased significantly in DS2205 compared to AlCoCrFeNi_2.1,_ with an increase in temperature from 450 °C to 650 °C. At 650 °C, the Cr depletion depth for AlCoCrFeNi_2.1_ was ~50 μm_,_ while that for DS2205 was ~300 μm. This suggests that Al in the AlCoCrFeNi_2.1_ acted as a sacrificial element and reduced Cr dissolution rate in the molten salt compared to DS2205. Redox control technique has been utilized for corrosion mitigation in molten fluoride and chloride salts [48]. Dissolution of Al and Cr into the molten salt resulted in the formation of AlCl_3_ and CrCl_2,_ which formed a redox couple of Al/AlCl_3_ and Cr/CrCl_2_. At the redox potential, Co, Ni, and Fe become relatively nobler and their dissolution is thermodynamically unfavorable. These findings suggest that Al/AlCl_3_ and Cr/CrCl_2_ couple may be used as redox buffer to slow or stop the dissolution of Fe, Co, and Ni from AlCoCrFeNi_2.1_ [49].

The chemical composition of an alloy determines the thermodynamic driving force for reaction between the molten salt and the alloy constituents [50]. Corrosion of an alloy in NaCl-KCl-MgCl_2_ molten salt is governed by the difference in Gibbs free energy of formation (ΔG°) for the salt constituents (NaCl, KCl, MgCl_2_) and corrosion products in the form of chlorides of the alloying elements (e.g., AlCl_3_, CrCl_2_, MnCl_2,_ SiCl_4_, etc). Corrosion products with lower values of ΔG° are more likely to form and the associated alloying elements are likely to preferentially react with the salt and selectively leach out from the alloy. This leads to thermodynamic driving force for selective depletion of the alloying elements, i.e., dealloying at the surface [49,51]. The interaction between chlorides and moisture impurity in the salt results in the formation of highly corrosive HCl and metal oxides/hydroxides [49,51]:(3)$$MCl\left(l\right)+H_{2}O\;\to MOH\left(l\right)+HCl\left(g\right)$$

The generated HCl reacts with the constituent elements of the alloy to form soluble metal chlorides as follows:(4)$$M_{e}+2HCl\left(l\right)\to M_{e}Cl_{2}+H_{2}\left(g\right)$$
where M_e_ is an alloying element (e.g., Al, Co, Cr, Fe, Mn, Si, etc.). Thermodynamic calculations were employed using FactSage (version 7.3, FactPS database, Montreal, Canada) to determine the potential reactions at high temperatures [48,51,52]. Based on ΔG° for formation of chloride corrosion products shown in Figure 7 (calculations at 450 °C and 650 °C), the alloying elements present in AlCoCrFeNi_2.1_ are expected to show high reactivity with the molten salt in the following order: Al > Cr > Fe > Co > Ni. This indicates that depletion of Al is the most favorable thermodynamically, leading to the formation of AlCl_3_. Similarly, for DS2205, alloying elements are expected to show high reactivity with the molten salt in the following order: Si > Mn > Cr > Fe > Ni > Mo. Thus, depletion of Si is most favorable thermodynamically in the case of DS2205, leading to the formation of SiCl_4_. Trace amount of salt impurities (Table 1) in NaCl-KCl-MgCl_2_ may also have contributed towards the relative dissolution rates of the alloying elements in the molten salt. 

Metallic chloride impurities (e.g., NiCl_2_, FeCl_2_, CoCl_2_), commonly present in NaCl-KCl-MgCl_2_ molten salt, may react with the alloying elements, leading to their dissolution in the melt. Ni, Co, and Fe impurity chlorides may cause depletion of Cr via the following substitution (oxidation-reduction) reactions [49,50]:(5)$$NiCl_{2}\left(l\right)+Cr\left(s\right)\to CrCl_{2}+Ni\left(s\right)$$
(6)$$FeCl_{2}\left(l\right)+Cr\left(s\right)\to CrCl_{2}+Fe\left(s\right)$$
(7)$$CoCl_{2}\left(l\right)+Cr\left(s\right)\to CrCl_{2}+Co\left(s\right)$$

Figure 6a,c shows that there is enrichment of Co and Ni (and to some degree Fe) in the near-surface region even more than the bulk alloy, suggesting that Co, Ni, and Fe may be enriched in the Al and Cr depleted regions. Ni^2+^, Co^2+^, and Fe^2+^ ions were reduced to atomic Ni, Co, and Fe at the surface and their diffusion was likely assisted by vacancies created during depletion of Al and Cr. However, similar effects were not observed in case of DS2205 duplex steel, possibly because of the lack of stable redox couple of Al/AlCl_3_ and Cr/CrCl_2_. In addition, there was significant depletion of the refractory element Mo from DS2205, as shown in Figure 6b,d. Figure 7 shows relatively high ΔG° for MoCl_2_, so depletion of Mo due to chloride formation is not expected. Hence, depletion of Mo in DS2205 may be caused by direct reaction with moisture and oxide impurities, resulting in the formation of unstable or volatile metallic oxides as follows [53]:(8)$$xMo\left(s\right)+yH_{2}O\left(g\right)\to Mo_{x}O_{y}+yH_{2}\left(g\right)$$

### 4.2. Effect of Alloy Microstructure on Molten Salt Corrosion Behavior

To explain the effect of eutectic microstructure on the molten salt corrosion behavior, scanning kelvin probe (SKP) was utilized to evaluate the relative work function (WF) behavior on the surface of the alloys [27,37]. The WF may be directly related to the ability for charge transfer and represents the relative nobility of a metal surface [54]. SKP potential measured in air for several metals was found to vary linearly with their corrosion potential in aqueous solution [36,55]. Phase-specific work function for the as-cast AlCoCrFeNi_2.1_ and DS2205 was measured to understand the micro-galvanic corrosion and to investigate the change in galvanic activity between L1_2_ and B2 in AlCoCrFeNi_2.1_ and between FCC and BCC in DS2205. Respective SKP maps are shown for DS2205 in Figure 8a and for AlCoCrFeNi_2.1_ in Figure 8b. 

The color contrast in the topographical plot indicates the relative work function potential contrast difference between the FCC and BCC phases. The FCC phase (red-colored areas) in AlCoCrFeNi_2.1_ (L1_2_) and DS2205 (FCC) showed higher relative work function compared to the corresponding BCC phase (green-blue-colored areas) in AlCoCrFeNi_2.1_ (B2) and DS2205 (BCC). The potential contrast difference between the phases in AlCoCrFeNi_2.1_ was approximately ~50 mV, while in the case of DS2205 it was ~35 mV, which explains the selective dissolution seen from the EDS maps, due to the micro-galvanic cell formation in both the alloys. As schematically shown in Figure 8c,d, the high relative work function of the L1_2_ and FCC phase indicates their higher nobility compared to the B2 and BCC phase, respectively, and agrees with the SEM-EDS data (Figure 5 and Figure 6) of selective corrosion of B2/BCC phase. Co-Cr-Fe-rich FCC-L12 may be considered the matrix in this alloy that triggers anodic dissolution of the neighboring Al-Ni-rich B2 phase. It has been reported previously that Cr-Mo-rich α—Ferrite (BCC) (Table 2) is considered anodic with respect to Ni-rich γ—Austenite (FCC) (Table 2), which is cathodic in DS2205. The overall relative work function potential of AlCoCrFeNi_2.1_ alloy (average ~50 mV) was higher compared to DS2205 alloy (average ~−65 mV), indicating the nobility of AlCoCrFeNi_2.1_ versus DS2205, and agrees with the lower corrosion rate observed in AlCoCrFeNi_2.1_ rather than DS2205 at both the temperatures.

## 5. Conclusions

The corrosion behavior of dual-phase AlCoCrFeNi_2.1_ (EHEA) and duplex stainless steel 2205 (DS2205) was studied in NaCl-KCl-MgCl_2_ molten salt at 450 °C and 650 °C. AlCoCrFeNi_2.1_ EHEA showed superior molten salt corrosion resistance than DS2205, which was attributed to the following: (i)Higher Ni content in EHEA and sacrificial role of Al in reducing outward diffusion of Cr, Fe, and Ni in AlCoCrFeNi_2.1_ and the different reactivity of formation of chlorides, as well as enrichment of Co and Ni in the corrosion layer.(ii)Thermodynamically driven corrosion between alloying elements and molten salt constituents or impurities present within the melt. Metallic impurities such as Ni^2+^, Fe^2+^, and Co^2+^ enhance the corrosion resistance of AlCoCrFeNi_2.1_ via enrichment of impurity ions such as Ni, Co, and Fe (to lesser extent) in the corrosion layer as temperature increases.(iii)A lower volume fraction of the BCC phase, which makes it less prone to galvanic corrosion. The phase-specific work function for the as-cast AlCoCrFeNi_2.1_ and DS2205 suggested micro-galvanic corrosion, with nobler L1_2_ and FCC phase in EHEA and DS 2205, respectively, compared to the B2 and BCC phases.

## Figures and Tables

**Figure 1 entropy-25-00296-f001:**
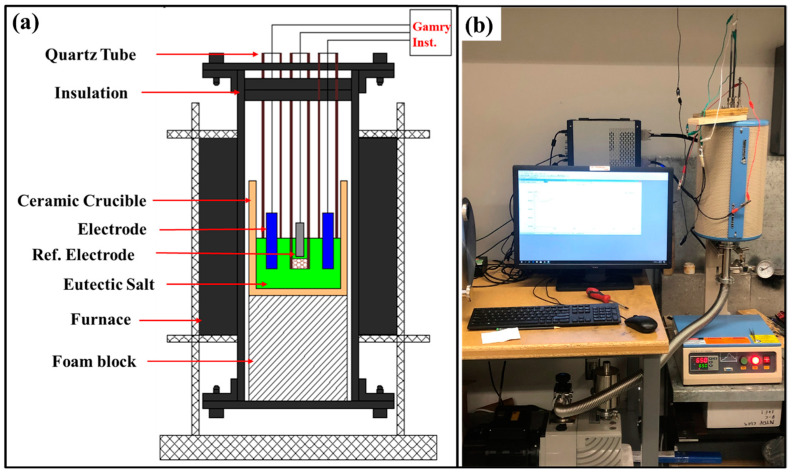
(**a**) Schematic illustration and (**b**) experimental set-up of three-electrode tube furnace used in the present study for corrosion test in molten NaCl-KCl-MgCl_2_ salt at 450 °C and 650 °C.

**Figure 2 entropy-25-00296-f002:**
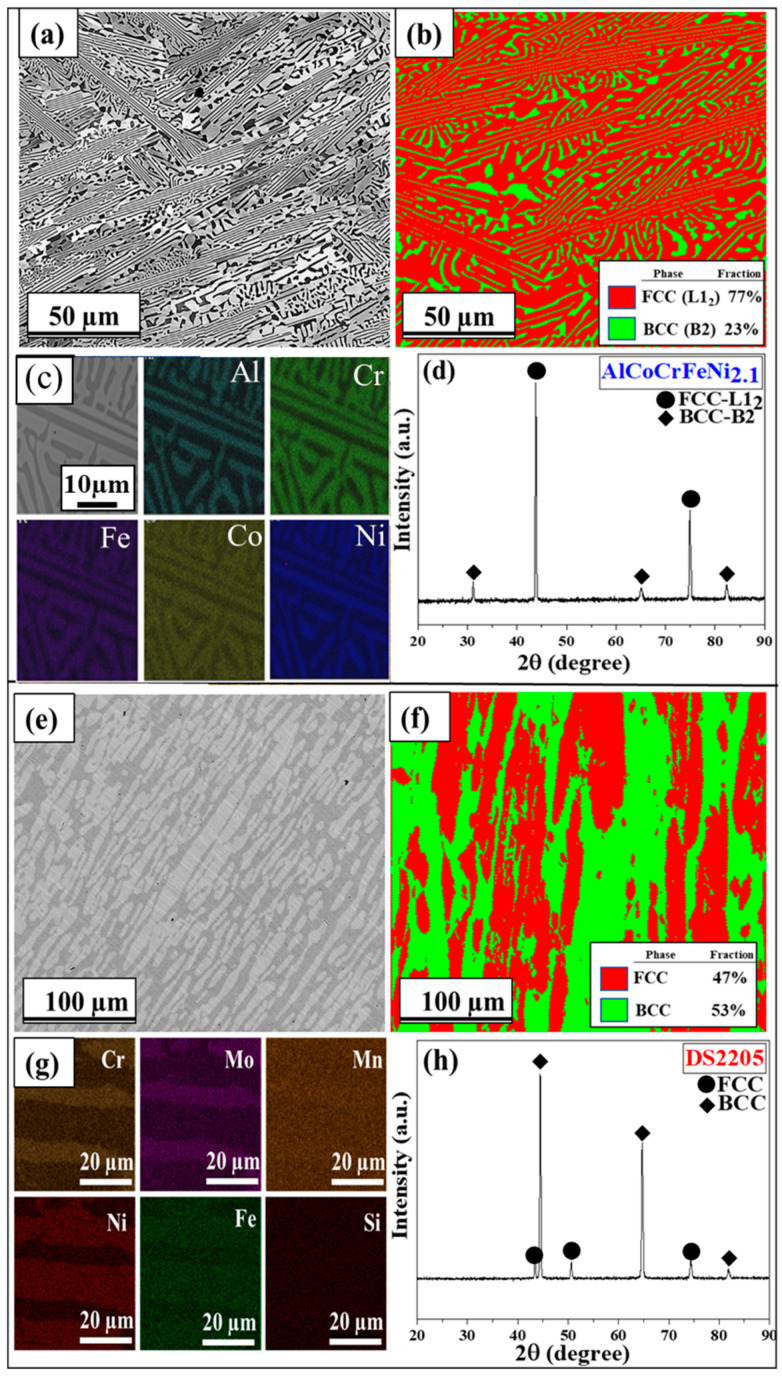
SEM BSE image for (**a**) EHEA and (**e**) DS2205; EBSD phase map for (**b**) EHEA and (**f**) DS2205; EDS map for (**c**) EHEA and (**g**) DS2205; XRD pattern for (**d**) EHEA and (**h**) DS2205.

**Figure 3 entropy-25-00296-f003:**
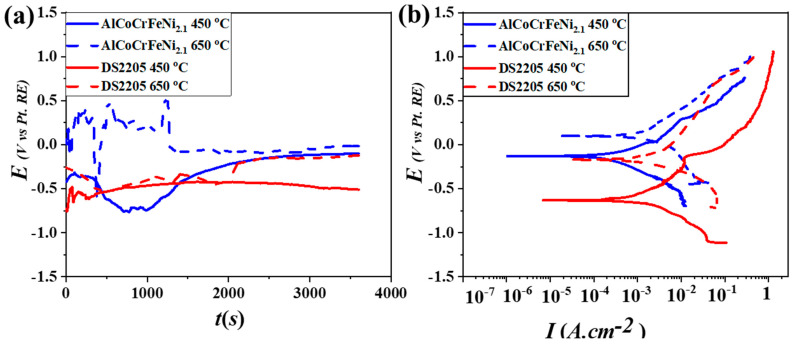
(**a**) OCP curves of the AlCoCrFeNi_2.1_ and DS2205 alloys exposed to chloride salt medium at 450 °C and 650 °C; (**b**) potentiodynamic polarization curves of the alloys at 450 °C and 650 °C, showing better corrosion resistance of AlCoCrFeNi_2.1_ at 450 °C and 650 °C compared to DS2205.

**Figure 4 entropy-25-00296-f004:**
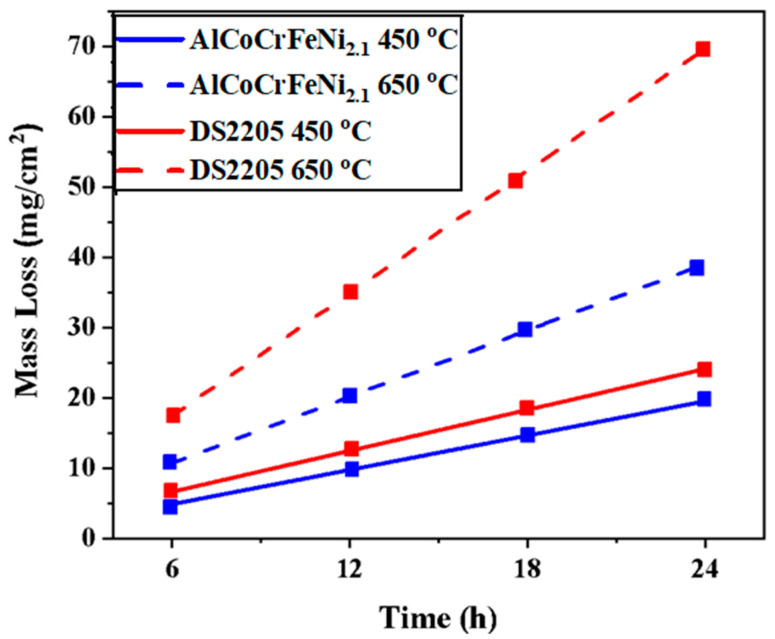
Mass loss of AlCoCrFeNi_2.1_ and DS2205 after exposure in NaCl-KCl-MgCl_2_ molten salt at 450 °C and 650 °C for 24 h.

**Figure 5 entropy-25-00296-f005:**
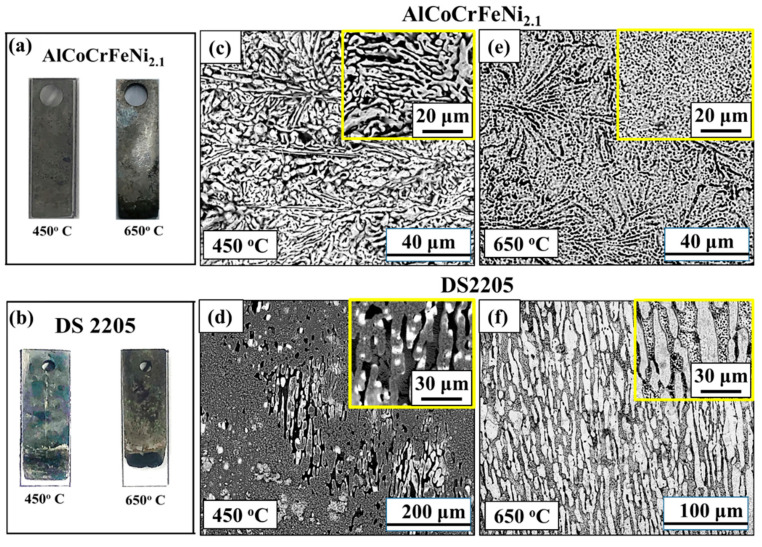
Optical images for (**a**) AlCoCrFeNi_2.1_ and (**b**) DS2205 after electrochemical corrosion experiments in molten NaCl-KCl-MgCl_2_ salt; SEM image of the corroded surface at 450 °C for (**c**) AlCoCrFeNi_2.1_ EHEA and (**d**) DS2205; SEM image of the corroded surface at 650 °C for (**e**) AlCoCrFeNi_2.1_ EHEA and (**f**) DS2205. The insets in figures (**c**) through (**f**) show zoomed-in view of the corroded surfaces indicating selective removal of B2 phase in AlCoCrFeNi_2.1_ and BCC phase in DS2205 because of galvanic coupling.

**Figure 6 entropy-25-00296-f006:**
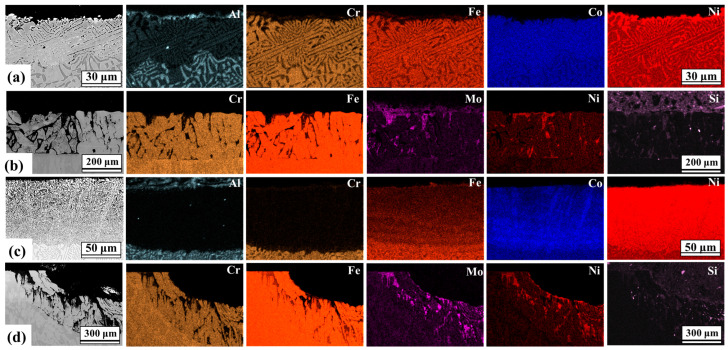
Cross section EDS maps of the near-surface composition of alloys exposed to molten NaCl-KCl-MgCl_2_ for (**a**) AlCoCrFeNi_2.1_ at 450 °C, (**b**) DS2205 at 450 °C, (**c**) AlCoCrFeNi_2.1_ at 650 °C, and (**d**) DS2205 at 650 °C.

**Figure 7 entropy-25-00296-f007:**
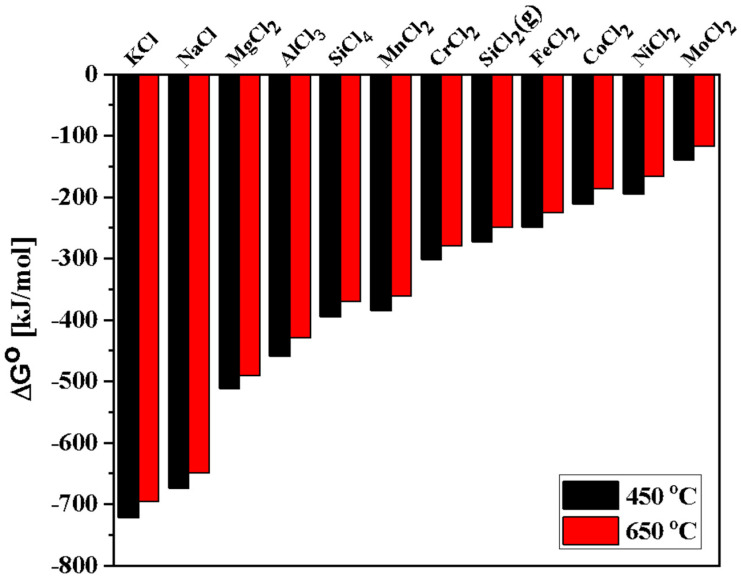
Gibbs free energies of the chemical reaction between the constituent elements and 1 mol Cl^−2^ as a function of temperature exposed to NaCl-KCl-MgCl_2_ under atmospheric air.

**Figure 8 entropy-25-00296-f008:**
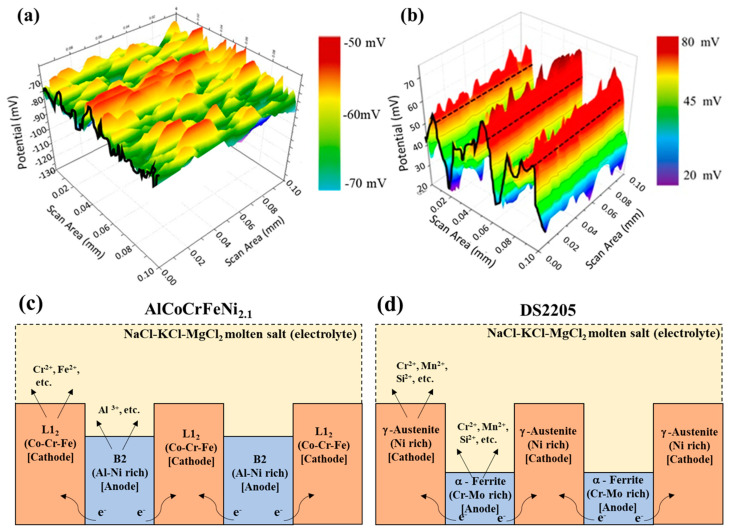
Relative work-function topography obtained from scanning kelvin probe microscope of the phases to delineate the potential changes in (**a**) as-cast DS2205 and (**b**) as-cast AlCoCrFeNi_2.1_. Schematic illustration of the mechanism of galvanic corrosion for (**c**) the AlCoCrFeNi_2.1_ alloy Ni-rich BCC-B2 phase in the vicinity of FCC-L1_2_; (**d**) the DS2205 alloy Cr-Mo-rich BCC phase in the vicinity of Ni-rich FCC phase.

**Table 1 entropy-25-00296-t001:** Concentration of main impurities in NaCl-KCl-MgCl_2_ eutectic salt before corrosion test (i.e., immediately following salt pre-conditioning) determined by ICP-OES.

	Cr [ppm]	Fe [ppm]	Ni [ppm]	Co [ppm]	Si [ppm]	Al [ppm]
NaCl-KCl-MgCl_2_	≤15	≤50	≤10	≤10	≤125	≤65

**Table 2 entropy-25-00296-t002:** Chemical composition (at. %) of the phases in AlCoCrFeNi_2.1_ and DS2205.

AlCoCrFeNi_2.1_ (at. %)				DS2205 (at. %)			
Element	FCC L1_2_	BCC B2	Nominal Comp.	Element	FCC γ—Austenite	BCC α—Ferrite	Nominal Comp.
**Al**	6.4 ± 0.3	15.7 ± 0.7	16.4	**Cr**	21.6 ± 0.7	24.8 ± 0.3	22.24
**Co**	18.4 ± 0.3	15.1 ± 0.2	16.4	**Mo**	2.1 ± 0.1	4.1 ± 0.2	3.18
**Cr**	23.1 ± 0.4	14.3 ± 0.3	16.4	**Ni**	6.9 ± 0.5	4.2 ± 0.3	5.75
**Fe**	20.5 ± 0.4	14.9 ± 0.2	16.4	**Mn**	1.9 ± 0.2	1.6 ± 0.4	1.86
**Ni**	31.6 ± 0.5	40.3 ± 0.7	34.4	**Si**	0.7 ± 0.2	0.5 ± 0.1	0.77
				**Fe**	64.3 ± 0.7	66.5 ± 0.6	Balance

**Table 3 entropy-25-00296-t003:** Corrosion potential (E_corr_), corrosion current density (I_corr_), and corrosion rate (CR) of AlCoCrFeNi_2.1_ and DS2205 alloys in NaCl-2KCl-MgCl_2_ salt at 450 °C and 650 °C.

	450 °C				650 °C			
	E_corr_ (V)	I_corr_ (A.cm^−2^)	CR (Equation (1)) (mm/year)	CR (Equation (2)) (mm/year)	E_corr_ (V)	I_corr_ (A.cm^−2^)	CR (Equation (1)) (mm/year)	CR (Equation (2)) (mm/year)
**AlCoCrFeNi_2.1_**	−0.26	0.17 × 10^−3^	1.18 ± 1	3.58	−0.13	1.4 × 10^−3^	9.2 ± 1	15.67
**DS2205**	−0.59	0.46 × 10^−3^	4.78 ± 1	9.5	−0.18	1.9 × 10^−3^	19.86 ± 1	32.84

## Data Availability

All the data obtained for this work are available upon request from the corresponding author.

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
