# Peer review of "Molten Salt Corrosion Behavior of Dual-Phase High Entropy Alloy for Concentrating Solar Power Systems"

_entropy, 2023, doi:10.3390/e25020296_

Round 1
Reviewer 1 Report
Comment:Major revision
This work was very interesting,but more evidence and further explanation are needed.
1. The use of chloride molten salt is designed to make the CSP more efficient at higher temperatures. So, the author should set the temperature at the range of 600-800 oC.
2. What is the selection basis of three electrode materials? It seems unreasonable to choose the same material for both WE and CE.
3. As to maximize the display of valid information, figure 2e and 2f are recommended not to overlap.
4. The content of elements in Table 2 seems to not be consistent with EDS map results (Fig. 2e-f).
5. Please give the explanation of curve changes at different stages in figure 3a.
6. It is suggested to redraw the graph and it is difficult to distinguish curves in figure 3b. The color of the label is inconsistent with the curves in figure 4.
7. It is recommended to calculate the annual corrosion rate according to weight gain curve and compare it with that obtained by corrosion current.
8. All analysis were based on chloride products. How to determine if the corrosion product is chloride not oxide? The elemental distribution did not contain Cl.
9. All the formula are not clear.
10. How to determine the phases in figure 8a-b?
Reviewer 2 Report
This submission is quite fine, however two points have to be revised:
1. Fig. 6 shows results but is presented in the discussion section.
2. In the context of Fig. 8, it is written twice: "Error! Reference source not found".
Reviewer 3 Report
The manuscript is very interesting and informative, the authors have done a lot of work. However, I have a number of comments and questions.
1) References [28] and [29] are duplicated, please correct.
2) Lines 119, 120, Table 2. Ni and Al have an FCC lattice, while Cr has a BCC lattice. For EHEA you write the other way around, why?
3) What is the basis for choosing temperatures for corrosion testing?
4) (Eq 1). Where did the numerical value of the constant K1 come from?
5) In Fig. 5 for DS2205 after 650°C the surface is less damaged than when tested at 450°C. How do you explain it?
6) In Fig. 6 in corrosion products there are only oxide compounds. Where did the oxygen in molten salts come from? Why didn't chlorides form?
Round 2
Reviewer 1 Report
Comment:It can be accepted in present form.
Author Response
It can be accepted in present form
Reviewer 3 Report
1. It is necessary to add to the text of the manuscript in the section "Materials and Methods" an explanation about the choice of temperatures for the study.
2. It is necessary to add data to the text of the manuscript from where the constant K1 in equation (1) is taken.
3. Are there chlorine distribution maps to add to Fig. 6?
Author Response
- It is necessary to add to the text of the manuscript in the section "Materials and Methods" an explanation about the choice of temperatures for the study.
Ans: Following texts were added to the Manuscript. "One application is in an indirect storage system with molten salt as a storage medium often used in a parabolic trough CSP plant with maximum temperatures in the range of around 400-450 °C. Another configuration is a direct storage system often used in a tower CSP plant with molten salt heat transfer fluid (HTF) as well as TES with maximum temperatures in the range of around 600-650 °C [4-7]".
2. It is necessary to add data to the text of the manuscript from where the constant K1 in equation (1) is taken.
Ans: K1 value from equation 1 is taken from ASTM G 102.
3. Are there chlorine distribution maps to add to Fig.6?
Ans: No, as depending on the solubility of these chloride species in the liquid chloride melt at the given temperature, they will not sustain and will eventually dissolve in the molten chloride salt. Due to that, there is no chlorine distribution maps in Fig.6.